Psychometric properties of the Swedish version of the satisfaction with life scale in a sample of individuals with mental illness

http://orcid.org/0000-0003-3356-4454 Garcia Danilo 1 2 3 4 5 danilo.garcia@icloud.com
Nima Ali Al 3 4 5
http://orcid.org/0000-0001-6639-9809 Kazemitabar Maryam 6 7
Amato Clara 5 8
Lucchese Franco 8 9
Mihailovic Marko 10 11
Kijima Nobuhiko 12 13
1 Department of Behavioral Sciences and Learning, Linköping University , Linköping , Sweden
2 Centre for Ethics, Law and Mental Health (CELAM), University of Gothenburg , Gothenburg , Sweden
3 Department of Psychology, University of Gothenburg , Gothenburg , Sweden
4 Promotion of Health and Innovation (PHI) Lab, Network for Well-Being , Sweden
5 Blekinge Centre of Competence, Region Blekinge , Karlskrona , Sweden
6 Department of Psychology, University of Tehran , Tehran , Iran
7 Promotion of Health and Innovation (PHI) Lab, Network for Well-Being , Iran
8 Promotion of Health and Innovation (PHI) Lab, Network for Well-Being , Italy
9 Department of Dynamic and Clinical Psychology, University of Rome “La Sapienza” , Rome , Italy
10 Department of Psychiatry and Behavioral Sciences, Northwestern University , Chicago , United States
11 Promotion of Health and Innovation (PHI) Lab, Network for Well-Being , United States
12 Promotion of Health and Innovation (PHI) Lab, Network for Well-Being , Japan
13 Faculty of Business and Commerce, Keio University , Tokyo , Japan
Palazón-Bru Antonio
Electronic publication date: 2021 May 12
Publication date: 2021
Volume: 9
Electronic Location ID: e11432
Received 2021 Jan 25; Accepted 2021 Apr 20
Copyright: © 2021 Garcia et al.
Copyright year: 2021
Copyright holder: Garcia et al.
License: This is an open access article distributed under the terms of the Creative Commons Attribution License, which permits unrestricted use, distribution, reproduction and adaptation in any medium and for any purpose provided that it is properly attributed. For attribution, the original author(s), title, publication source (PeerJ) and either DOI or URL of the article must be cited.
License URL: https://creativecommons.org/licenses/by/4.0/

Keywords: Item response theory, Life satisfaction, Satisfaction with life scale, Mental illness, Classical test theory

Funding: Region Blekinge and the five Municipalities of Blekinge (Sölvesborg, Olofström, Karlshamn, Ronneby, and Karlskrona) The project was supported by Region Blekinge and the five Municipalities of Blekinge (Sölvesborg, Olofström, Karlshamn, Ronneby, and Karlskrona) thought their Research and Development agreement (i.e., FoU-avtalet). The funders had no role in study design, data collection and analysis, decision to publish, or preparation of the manuscript.

==============================
Background

Health assessment among individuals with mental health problems often involves measures of ill-being (e.g., anxiety, depression). Health is, however, defined as a state of physical, mental and social well-being and not merely the absence of disease (WHO, 1948, 2001). Hence, in order to address mental illness during the 21st century, we need to develop methods for the prevention, identification and treatment of mental illness; but also, for the promotion, identification, and maintenance of well-being. In this context, over three decades of subjective well-being research have resulted in the development of measures of positive aspects of human life, such as the Satisfaction with Life Scale (Diener et al., 1985). Our aim was to investigate the psychometric properties of the Satisfaction with Life Scale in a Swedish population of individuals with mental illness using both Classical Test Theory (CTT) and Item Response Theory (IRT).

Method

A total of 264 participants (age mean = 43.46, SD = 13.31) diagnosed with different types of mental illness answered to the Swedish version of the Satisfaction with Life Scale (five items, 7-point scale: 1 = strongly disagree, 7 = strongly agree).

Results

We found positive and significant relationships between the five items of the scale (r ranging from 0.37 to 0.75), good reliability (Cronbach’s alpha = 0.86), and that the one-factor solution had best goodness of fit (loadings between 0.52–0.88, p < 0.001). Additionally, there were no significant differences in comparative fit indexes regarding gender and occupation status. All items had high discrimination values (between 1.95–3.81), but item 5 (“If I could live my life over, I would change almost nothing”); which had a moderate discrimination value (1.17) and the highest estimated difficulty on response 7 (3.06). Moreover, item 2 (“The conditions of my life are excellent”) had less discrimination and redundant difficulty with both item 1 (“In most ways my life is close to my ideal”; 2.03) on response 7 and with item 3 (“I am satisfied with my life”; –1.21) on response 1. The five items together provided good information, with especial good reliability and small standard error within −1.00 up to about 2.00 and the highest amount of test information at 0.00 of the level of life satisfaction within this population.

Conclusions

Consistent with previous research, the scale had good reliability and provided good information across most of the latent trait range. In addition, within this population, sociodemographic factors such as gender and occupation status do not influence how individuals respond to the items in the scale. However, the items couldn’t measure extreme levels of low/high life satisfaction. We suggest replication of these findings, the test of additional items, and the modification of items 2 and 5 in order to use the scale among individuals with mental illness.

Introduction

About half of the proportion of people on sick leave in Sweden are due to mental illness. This trend has been steadily increasing for the past 10 years. The term mental illness has been, however, difficult to define. One part of the definition encompasses the fact that about 90% of those who receive this diagnosis have one or more common mental disorders (Vingård, 2020). Common mental disorders include depression, general anxiety disorder, panic disorder, post-traumatic stress disorder, some phobic disorders, and obsessive-compulsive disorder. Indeed, among individuals with mental health problems, health assessment often involves measures of ill-being (e.g., anxiety, depression, stress). Health is, however, defined as a state of physical, mental and social well-being and not merely the absence of disease (WHO, 1948, 2001). Hence, in order to address mental illness during the 21st century, one of the challenges is to develop methods for the prevention, identification and treatment of mental illness; but also, for the promotion, identification, and maintenance of well-being (Garcia et al., 2021).

Fortunately, over three decades of well-being research have resulted in the development of measures of positive aspects of human life, such as, subjective well-being (e.g., Diener, 1984, 2000; Diener, Lucas & Oishi, 2018; Diener et al., 2010). Since these measures are mostly self-reports, a cornerstone in their implementation is the testing of their psychometric properties in different populations (Pavot, 2018). One of the most prominent and commonly used instruments to operationalize the cognitive component of subjective well-being, the Satisfaction with Life Scale, measures people’s evaluations of life as a whole in relation to a psychological self-imposed ideal (Diener et al., 1985; Pavot & Diener, 1993; Pavot & Diener, 2008; Pavot et al., 1991). The psychometric properties of the Satisfaction with Life Scale have been validated in different populations, mostly using Classical Test Theory (CTT) (Chinni & Hubley, 2014; Emerson, Guhn & Gadermann, 2017; Nima et al., 2020a). Nevertheless, despite the usefulness of common CTT techniques, the results are dependent on the characteristics of the sample and the scale, thus, current research needs to address the psychometric properties of the Satisfaction with Life Scale using complementary methods, such as, Item Response Theory (IRT; Oishi, 2006, 2007; Nima et al., 2020b).

More specifically, changes in the characteristics of the sample (e.g., sample size, gender, and other sociodemographic factors) might have a significant effect on the psychometric properties of the measure at both item and scale levels (Oishi, 2007, 2006). For instance, individuals with mental illness do not only differ from the general population with regard to sociodemographic factors, such as, employment (Marwaha et al., 2007), (un)employment varies within this population (Granjard et al., 2021b). Additionally, CTT methodology does not provide researchers with sufficient information at different points along the scales’ continuum (e.g., ranging from extremely satisfied with life to extremely unsatisfied with life), because in CTT the scale is assumed to have equal measurement problems or capacity at all points of the scales’ continuum (Oishi, 2007). However, some items might be of varying weigh in relation to a person’s life satisfaction and therefore influence the participant’s actual score in life satisfaction in different ways (Oishi, 2007). IRT methods, in contrast, provide information about how each item helps to identify individuals with different levels of life satisfaction and also at which specific levels within the specific item this identification can be measured (Oishi, 2007). In other words, IRT methods depend basically on the assumption that individuals are independent of one another and that items behave in the same way for all individuals, which means that the precision of location estimates pertain, besides the group’s level of life satisfaction, the individual’s own level as well. This is in contrast to CTT, where the person’s test score is dependent on the items of the specific test and where the items’ properties (e.g., difficulty and discrimination) in turn, are dependent on the characteristics of the sample. Hence, IRT methods are sample and test independent (Oishi, 2007; see also Kijima et al., 1998).

With regard to the Satisfaction with Life Scale, CTT studies have replicated the single factor structure of the scale in different populations (Chinni & Hubley, 2014; Emerson, Guhn & Gadermann, 2017). Nevertheless, both CTT and the few studies using IRT methodology show that item 5 (“If I could live my life over, I would change almost nothing”) is less accurate for the measurement of life satisfaction compared to the other four (Pavot & Diener, 2008; Oishi, 2006, 2007; Vittersø, Biswas-Diener & Diener, 2005; Nima et al., 2020a, 2020b; for the same results among individuals with mental illness, see Jovanovic, Lazic & Gavrilov-Jerkovic, 2020). This specific item represents an evaluation over one’s past life, thus, it has been suggested that it probably requires respondents to make more of a global recollection compared to, for example, item 2 (e.g., “The conditions of my life are excellent”), which rather requires the apprehension of one’s present life (Pavot & Diener, 2008). That being said, since responses to item 5 are significantly correlated with the responses to the other four items, researchers are usually recommended to keep it in the scale (Pavot & Diener, 2008). That being said, as accurately suggested by Jovanovic, Lazic & Gavrilov-Jerkovic (2020), individuals with mental illness have memory issues regarding the past (e.g., negativity bias), thus, such temporal-related problems with specific items need to be scrutinized using IRT among populations with mental illness.

In sum, when researchers validate subjective well-being measures, many of the disadvantages of CTT and the advantages of IRT have been neglected (Oishi, 2007). In addition, while many studies have used the Satisfaction with Life Scale to measure subjective well-being among individuals with mental illness (e.g., Meyer et al., 2004), only a few studies have addressed the psychometric properties of the scale in this population (e.g., Arrindell, Meeuwesen & Huyse, 1991; Jovanovic, Lazic & Gavrilov-Jerkovic, 2020). Therefore, in order to propose this scale as a possible measure of subjective well-being among individuals with mental illness, our aim was to investigate the psychometric properties of the Satisfaction with Life Scale in a Swedish population of individuals with mental illness using both CTT and IRT. More specifically, we tested the expected positive significant intercorrelations between the five items in the scale (1); we explored (2) and confirmed (3) the expected single factor structure of life satisfaction; we tested if the scale measured life satisfaction equally across both gender and occupation status groups (4); and finally, we tested the five items’ information characteristics using IRT (5). See Table 1 for the Swedish version of the scale.

Table 1 The Swedish version (Garcia & Siddiqui, 2009a, 2009b) of the Satisfaction with Life Scale (Diener et al., 1985).

	English	Swedish	
Instructions	Using the 1–7 scale below, indicate your agreement with each item by placing the appropriate number on the line preceding that item. Please be open and honest in your responding.	Använd den sjugradiga skalan nedanför för att ange ditt förhållningssätt till varje påstående genom att skriva lämplig siffra på raden framför.	
Item 1	In most ways my life is close to my ideal.	I de flesta avseende är mitt liv nära mitt ideal.	
Item 2	The conditions of my life are excellent.	Förutsättningarna i mitt liv är utmärkta.	
Item 3	I am satisfied with my life.	Jag är nöjd med mitt liv.	
Item 4	So far, I have gotten the important things I want in life.	Än så länge har jag fått de viktiga sakerna i livet jag vill ha.	
Item 5	If I could live my life over, I would change almost nothing.	Om jag kunde leva om mitt liv skulle jag i stort sett inte ändra på något alls.	
Note:

The Satisfaction with Life Scale from Diener et al. (1985). The satisfaction with life scale. Journal of Personality Assessment, 49, 71–75. Translation to Swedish by Patricia Rosenberg and Johanna Ekberg. The seven-point Likert scale used is as follows [Swedish translation in brackets]: 1 = Strongly disagree [Håller verkligen inte med]; 2 = Disagree [Håller inte med]; 3 = Slightly disagree [Håller till viss del inte med]; 4 = Neither agree nor disagree [Håller varken med eller inte med]; 5 = Slightly agree [Håller med till viss del]; 6 = Agree [Håller med]; 7 = Strongly agree [Håller verkligen med].

Method

Ethical statement

The study (protocol 2017/7) was approved by the Swedish Ethical Review Authority (Dnr. 2017/305) and conducted in accordance with the ethical standards of the 1964 Helsinki declaration and further amendments. Hence, all participants were provided with the necessary information to obtain verbal consent (e.g., aims of the study, that participation was anonymous and voluntary).

Participants and procedure

The data used here has been previously published elsewhere (Granjard et al., 2021b), but it has never been analyzed as in the present study. All individuals with mental illness between 18–65 years of age who received support in each of the five municipalities in Blekinge were contacted (N = 621). The support includes: help with everyday finances, help with daily shores, transport, support when contacting authorities, help taking social contact, support seeking job or occupation, and help with lifestyle habits (https://sweden.se/society/swedens-disability-policy/). The survey was conducted at the outpatient clinics. The staff working closest to the clients were responsible for the exclusion procedure. A total of 146 individuals were excluded due to severe dementia or substance use disorder at the time the data was collected. Another 62 individuals declined to participate in the study and 126 individuals did not respond to the survey questions (i.e., a total 188 individuals who dropped out). Thus, the final sample represented roughly 60% of those eligible to participate (n = 475): 287 individuals (148 males, 134 females, and 5 missing) with a mean age of 43.46 years (SD = 13.32). Of these 287 individuals, 67% reported doing it without any help, 24% with some help, and 9% did not answered to this question. Regarding education, about 3.5% of the respondents did not finish primary school, 23.7% finished primary school, 52.3% had a high-school degree, 12.2% had higher education and 8.3% had other type of education. About 60.4% reported having an occupation. Most of the participants were single (76.3%) and lived in their own accommodation (74.6%) and only a few of them reported living in an institution for individuals with mental illness (16.4%). Fifteen participants out of the 287 respondents did not answer one or more of the five items in the Satisfaction with Life Scale. Since the missing data was ≤5%, we found it appropriate to use the listwise deletion method to handle the missing data. This method excludes respondents with missing scores on any variable or variables used in subsequent analyses. Thus, we ended up with a sample of 264 participants (age mean = 43.46, SD = 13.31, ranging from 17 to74 years of age) for the present study (see Fig. 1).

Figure 1 Recruitment procedure in the present study from the project “Brukarundersökning” (BrUS) conducted at the Center of Competence, Region Blekinge.

Instrument

The Satisfaction with Life Scale (Diener et al., 1985) assesses the cognitive component of subjective well-being and consists of 5 items (i.e., “In most ways my life is close to my ideal”, “The conditions of my life are excellent”, “I am satisfied with my life”, “So far I have gotten the important things I want in life”, and “If I could live my life over, I would change almost nothing”) that require a response on a 7-point Likert scale (1 = “strongly disagree”, 7 = “strongly agree”). The Swedish version of the Satisfaction with Scale (see Table 1) has been used in several studies in the Swedish normal population (e.g., Garcia & Siddiqui, 2009a, 2009b). The sum of the five items is expected to measure life satisfaction as a single factor.

Statistical procedure

Firstly, a Pearson correlations analyses was conducted to investigate the relationship between the five items of the Satisfaction with Life Scale. Secondly, we applied an exploratory factor analysis using principal components analysis in SPSS (version 24) to investigate if the items in the Swedish version of the Satisfaction with Life Scale loaded as one single factor, as suggested by previous research in other populations, in our sample of individuals with mental illness. Since there was only one latent factor, namely life satisfaction, we did not use any rotation method. Next, we conducted a confirmatory factor analysis using Structural Equation Modeling (SEM) in AMOS (version 24). In this analysis, we used the Maximum Likelihood estimation method to calculate fit indices and factor loadings. We used multi-group confirmatory factor analysis with three invariance models (i.e., configural, metric, and scalar) in order to test measurement invariance with regard to gender (females vs. males) and occupation status (employed vs. unemployed)—the only two sociodemographic categories with large enough subsamples for valid and reliable testing (cf. Muthén & Muthén, 2002; Kline, 2015). Finally, since the items in the Satisfaction with Life Scale are ordinal and scored on a Likert scale, we used Graded Response Model as the IRT technique in the last part of our analyses.

Results

Descriptive and correlational analysis

The results showed that there were positive and significant relationships among all the items in the scale. See Table 2 for the correlation coefficients, mean values, and standard deviations for all five items of the Swedish version of the Satisfaction with Life Scale.

Table 2 Correlations, means and standard deviations (±) for all five items of the Swedish version of the Satisfaction with Life Scale in a sample of individuals with mental illness.

ITEMS	SWLS1	SWLS2	SWLS3	SWLS4	SWLS5	
SWLS1	–					
SWLS2	0.72**	–				
SWLS3	0.75**	0.62**	–			
SWLS4	0.56**	0.55**	0.63**	–		
SWLS5	0.43**	0.37**	0.46**	0.43**	–	
Mean and SD	3.17 ± 1.78	3.44 ± 1.80	3.67 ± 1.80	3.85 ± 1.90	2.83 ± 1.91	
Notes:

** p < 0.001.

SWLS1: “In most ways my life is close to my ideal”, SWLS2: “The conditions of my life are excellent”, SWLS3: “I am satisfied with my life”, SWLS4: “So far, I have gotten the important things I want in life”, SWLS5: “If I could live my life over, I would change almost nothing”.

Exploratory factor analysis

We found only one latent factor with an eigenvalue higher that 1 (i.e., 3.23; cf. Tabachnick & Fidell, 2012. See Fig. 2). This latent factor explained 64.58% of the variance in participants’ life satisfaction. The loadings for each of the five items on the latent factor (i.e., life satisfaction) were: 0.88 for item 1 (“In most ways my life is close to my ideal”), 0.82 for item 2 (“The conditions of my life are excellent”), 0.88 for item 3 (“I am satisfied with my life”), 0.79 for item 4 (“So far I have gotten the important things I want in life”), and 0.63 for item 5 (“If I could live my life over, I would change almost nothing”). Finally, the scale had a Cronbach’s α = 0.86 in the present study. In sum, the results were acceptable, consistent with previous research, and suggested that the Satisfaction with Life Scale measures a single factor of life satisfaction in our sample of Swedish individuals with mental illness.

Figure 2 Scree plot for the principal component analysis of the Swedish version of the Satisfaction with Life Scale in a sample of individuals with mental illness.

Confirmatory factor analysis

The analysis showed that the chi-square value was significant (Chi2 = 22.57, df = 5, p < 0.001). That being said, the chi-square statistic is heavily influenced by sample size, thus, large samples have a higher likelihood of being significant (Kline, 2015). Since the other fit indices suggested a good model (comparative fit index = 0.97, incremental fit index = 0.97, and normed fit index = 0.97), we considered that the proposed unidimensional model was acceptable. All regression loadings between life satisfaction and the five items were significant at p < 0.001 and ranged from 0.52 to 0.88 (See Fig. 3).

Figure 3 Structural equation model of the Swedish version of the Satisfaction with Life Scale in a sample of individuals with mental illness. Showing all paths from the latent factor to the five items and their standardized parameter estimates.

Note: Chi-square = 22.57; df = 5; comparative fit index = 0.97; incremental fit index = 0.97; normed fit index = 0.97. e = error. N = 264. SWLS1: “In most ways my life is close to my ideal”, SWLS2: “The conditions of my life are excellent”, SWLS3: “I am satisfied with my life”, SWLS4: “So far, I have gotten the important things I want in life”, SWLS5: “If I could live my life over, I would change almost nothing”.

Measurement invariance

All the invariance models (i.e., configural, metric, and scalar) indicated no differences in responses to the Satisfaction with Life Scale across gender and occupation groups. More specifically, the difference in comparative fit indexes (CFIs) between the configural and metric model were less than 0.01. Importantly, since the configural model compares the overall factor structure of the two subsamples in each sociodemographic category group (Lee, 2018), thus, there was no difference in how females and males with mental illness and in how employed and unemployed individuals with mental illness responded to the items in the Satisfaction with Life Scales (Tables 3 and 4). Furthermore, the differences between the scalar model (intercepts) against the metric model (factor loadings) were not significant for any of the two sociodemographic categories (Tables 3 and 4). In other words, when comparing factor loadings and intercepts, there were no differences in responses to the Satisfaction with Life Scale’s items between males and females nor between employed and unemployed within this population of individuals with mental illness. In addition, the differences in CFIs between the metric and scalar models were less than 0.01, which also indicates that there were no differences in responses to the scale with regard to these two sociodemographic categories.

Table 3 Test of measurement invariance for gender.

Model	Chi-square	Degrees of Freedom	P-value	CFI	RMSEA	
Configural	32.378	10	0.0003	0.965	0.129	
Metric	35.264	14	0.0013	0.966	0.106	
Scalar	39.089	18	0.0028	0.967	0.094	
Models Compared	Chi-square	Degrees of Freedom	P-value	
Metric vs. Configural	2.887	4	0.5770	
Scalar vs. Configural	6.711	8	0.5681	
Scalar vs. Metric	3.825	4	0.4302	

Table 4 Test of measurement invariance for occupation status.

Model	Chi-square	Degrees of Freedom	P-value	CFI	RMSEA	
Configural	37.480	10	0.0000	0.957	0.142	
Metric	39.486	14	0.0003	0.960	0.116	
Scalar	45.411	18	0.0004	0.957	0.106	
Models Compared	Chi-square	Degrees of Freedom	P-value	
Metric against Configural	2.006	4	0.7346	
Scalar against Configural	7.930	8	0.4403	
Scalar against Metric	5.924	4	0.2049	

Graded response model

Regarding item discrimination, the results showed that all items had high discrimination values (Alpha, α, from 1.95 to 3.81). The only exception was item 5 (“If I could live my life over, I would change almost nothing”), which had moderate discrimination values (1.17). Items 1 to 4 showed a steeper slope, indicating that these items had good discrimination and can differentiate between persons with high and low levels of the latent score of life satisfaction better than item 5 (see Table 5). Regarding item difficulty, we found that item 5 had the highest estimated difficulty parameter (−0.66) and that item 4 (“So far I have gotten the important things I want in life”) had the lowest estimated difficulty parameter (−1.40). For example, for item 1 (“In most ways my life is close to my ideal”), a person with a score of −0.94 has a 50% chance of answering 1 (strongly disagree) rather than responses 2, 3, 4, 5, 6 or 7; a person with a score of −0.15 has a 50% chance of responding 1 or 2, rather than 3, 4, 5, 6 or 7; while a person with a score of 2.03 has a 50% chance of responding 7, rather than 1, 2, 3, 4, 5 or 6.

Table 5 Item response analysis of the Swedish version of the Satisfaction with Life Scale in a sample of individuals with mental illness.

Item	Parameter	Coef.	SE	Z	P	CI	95%	
SWLS1	Discrimination	3.81	0.50	7.70	0.00	2.84	4.79	
Difficulty							
>=2	−0.94	0.10	−9.32	0.00	−1.14	−0.74	
>=3	−0.15	0.08	−1.77	0.08	−0.31	0.02	
>=4	0.25	0.08	2.91	0.00	0.08	0.41	
>=5	0.64	0.09	7.00	0.00	0.46	0.82	
>=6	1.32	0.12	11.43	0.00	1.10	1.55	
7	2.03	0.18	11.43	0.00	1.68	2.38	
SWLS2	Discrimination	2.41	0.26	9.29	0.00	1.90	2.92	
Difficulty							
>=2	−1.21	0.13	−9.33	0.00	−1.47	−0.96	
>=3	−0.29	0.09	−3.05	0.00	−0.47	−0.10	
>=4	0.03	0.09	0.30	0.77	−0.15	0.21	
>=5	0.57	0.10	5.73	0.00	0.38	0.77	
>=6	1.30	0.13	9.82	0.00	1.04	1.57	
7	2.03	0.19	10.46	0.00	1.65	2.41	
SWLS3	Discrimination	3.23	0.37	8.68	0.00	2.50	3.96	
Difficulty							
>=2	−1.24	0.12	−10.36	0.00	−1.48	−1.01	
>=3	−0.52	0.09	−5.83	0.00	−0.70	−0.35	
>=4	−0.10	0.08	−1.14	0.26	−0.26	0.07	
>=5	0.40	0.09	4.51	0.00	0.23	0.58	
>=6	1.00	0.11	9.39	0.00	0.79	1.21	
7	1.82	0.16	11.40	0.00	1.51	2.13	
SWLS4	Discrimination	1.95	0.21	9.18	0.00	1.53	2.36	
Difficulty							
>=2	−1.40	0.16	−8.97	0.00	−1.70	−1.09	
>=3	−0.65	0.11	−5.82	0.00	−0.87	−0.43	
>=4	−0.17	0.10	−1.73	0.08	−0.37	0.02	
>=5	0.21	0.10	2.07	0.04	0.01	0.41	
>=6	0.94	0.13	7.47	0.00	0.69	1.18	
7	1.85	0.19	9.76	0.00	1.48	2.22	
SWLS5	Discrimination	1.17	0.16	7.37	0.00	0.86	1.48	
Difficulty							
>=2	−0.66	0.15	−4.24	0.00	−0.96	−0.35	
>=3	0.36	0.14	2.65	0.01	0.09	0.63	
>=4	0.73	0.15	4.75	0.00	0.43	1.02	
>=5	1.34	0.20	6.76	0.00	0.95	1.72	
>=6	2.07	0.27	7.58	0.00	1.53	2.60	
7	3.06	0.41	7.38	0.00	2.25	3.87	
Notes:

SWLS1: “In most ways my life is close to my ideal”, SWLS2: “The conditions of my life are excellent”, SWLS3: “I am satisfied with my life”, SWLS4: “So far, I have gotten the important things I want in life”, SWLS5: “If I could live my life over, I would change almost nothing”.

We graphed the item information function for each item to see how much information each item provided and to see at what level of the continuum each item had the most or least information. In other words, the item information function reflects the properties of each item in terms of both its difficulty and discrimination index. Here, item 1 (“In most ways my life is close to my ideal”) and item 3 (“I am satisfied with my life”) had the two highest discrimination estimates and provided more information than any of the remaining items (see Fig. 4). Finally, the test information function for the whole scale investigated how reliable the Satisfaction with Life Scale was. As shown in Fig. 5, the five items together provided information ranging between −1.00 and 2.00 to measure life satisfaction in our sample of Swedish individuals with mental illness. Thus, the scale had good reliability and a small standard error within this range. The highest amount of information and smallest standard error was at Theta = 0.00. However, there was almost no reliable information below −1.8 and above 2.5, that is, the standard error highly increases for both smaller and larger Theta values.

Figure 4 Boundary characteristic curves for the five items (A–E) of the Swedish version of the Satisfaction with Life Scale in a sample of individuals with mental illness.

Note: SWLS1: “In most ways my life is close to my ideal”, SWLS2: “The conditions of my life are excellent”, SWLS3: “I am satisfied with my life”, SWLS4: “So far, I have gotten the important things I want in life”, SWLS5: “If I could live my life over, I would change almost nothing”.

Figure 5 Items information function graphs for graded response for items in the Swedish version of the Satisfaction with Life Scale (A) and information and standard error graph for graded response (B) for the whole score of the Swedish version of the Satisfaction.

Note: SWLS1: “In most ways my life is close to my ideal”, SWLS2: “The conditions of my life are excellent”, SWLS3: “I am satisfied with my life”, SWLS4: “So far, I have gotten the important things I want in life”, SWLS5: “If I could live my life over, I would change almost nothing”.

Discussion

The current study examined, using both CTT and IRT, the psychometric properties of the Satisfaction with Life Scale in a Swedish sample of individuals with mental illness. Aligned with studies among other populations, our results supported the unidimensional structure of the Satisfaction with Life Scale (e.g., Neto, 1993; Oishi, 2006; Sachs, 2003) and the positive and significant relationships between the five items in the scale. More specifically, the scale had good reliability and best goodness of fit referring to a one single factor with significant loadings between the scale and items ranging from 0.52 to 0.88. Moreover, the scale measured life satisfaction across gender and occupational status in the same manner. This is also in line with findings among females and males in Croatian (Brdar, Anić & Rijavec, 2011) and Swedish samples (Hultell & Gustavsson, 2008). Our findings, however, add a new dimension by establishing that the scale measures life satisfaction equally among individuals with mental illness with and without occupation. This is important, since the ability and possibility of having an occupation varies largely in this population (Granjard et al., 2021b).

The IRT analyses showed that the items had high discrimination values. The only exception was item 5 (“If I could live my life over, I would change almost nothing.”), which had moderate discrimination value and the highest estimated difficulty on response 7. In contrast, item 4 (“So far I have gotten the important things I want in life.”) had the lowest estimated difficulty on response 1. Item 2 (“The conditions of my life are excellent”) had less discrimination and redundant difficulty with both item 1 (“In most ways my life is close to my ideal.”) on response 7 and with item 3 (“I am satisfied with my life.”) on response 1. Using item information function curves, we found that items 1 (“In most ways my life is close to my ideal.”) and 3 (“I am satisfied with my life.”) had the highest discrimination estimates and provided more information than the other items, while item 5 provided the less. Indeed, the few IRT studies conducted among general populations in China (Oishi, 2006) and the US (Nima et al., 2020b) have also shown that item 5 had the lowest discrimination estimate among all five items. As accurately pointed out by Pavot & Diener (2008), item 5 represents an evaluation over one’s past life, thus, it probably requires respondents to make more of a global recollection compared to, for example, item 1 (e.g., “In most ways my life is close to my ideal”), which rather requires the apprehension of one’s present life. Nevertheless, as accurately suggested by Jovanovic, Lazic & Gavrilov-Jerkovic (2020), individuals with mental illness have memory issues regarding the past (e.g., negativity bias), thus, such item-related temporal problems need to be scrutinized using IRT among populations with mental illness.

Nevertheless, these five items together provided good information, with especial good reliability and small standard error within −1.00 up to about 2.00 and the highest amount of test information at 0.00 of the level of life satisfaction within this population of individuals with mental illness. Moreover, comparative fit index, incremental fit index, and normed fit index were equal to 0.97 suggesting a good model fit for the Satisfaction with Life Scale among Swedish individuals with mental illness. These findings are also consistent with one study conducted in Iran among psychiatric outpatients. Indeed, the Persian version of the Satisfaction with Life Scale had acceptable internal consistency (a = 0.75), test–retest reliability (ICC = 0.64), comparative fit index (0.91), and root mean square error of approximation (0.05; Fallahi Khesht Masjedi & Pasandideh, 2016). In accordance, the validation of the Dutch version of the Satisfaction with Life Scale among a sample of psychiatric patients with severe mental illness also indicated acceptable loadings (>0.40), good internal consistency (a = 0.80), and fairly well corrected item-total correlations ranging from +0.40 to +0.70 (Arrindell, van Nieuwenhuizen & Luteijn, 2001). Finally, the Malaysian version of the Satisfaction with Life Scale among psychiatric and medical outpatients also showed very high goodness of fit indices (Chi-square/df = 1.108; GFI = 0.993; CFI = 0.999; RMSEA = 0.019) (Aishvarya et al., 2014). In sum, our results using CTT in a Swedish population of individuals with mental illness, replicate results showing that the Satisfaction with Life Scale has good validity and reliability among psychiatric patients in different cultures. Last but not the least, our IRT results are an important new addition to the scale’s validity at the item level in this specific population.

Limitations and suggestions for future research

Since our sample was relatively small we were not able to investigate measurement invariance within the group across sociodemographic factors, such as, type of psychiatric pathologies (cf. Boncori et al., 2011), age, marital status, and education. After all, we needed about 100–150 participants per group in order to conduct such analyses (Muthén & Muthén, 2002; Kline, 2015). We recommend that future studies compare life satisfaction as measured by the Satisfaction with Life Scale among individuals with mental illness and individuals without mental illness from the general Swedish population. This will help us to understand whether the Satisfaction with Life Scale is invariant among these two distinct populations. Another major limitation was that we were not able to have access in detail to all different diagnosis in this population. Mental illness is after all a wide concept (Vingård, 2020). As stated earlier, the answers to the items in the Satisfaction with Life Scale could differ between patients with different mental disorders. In addition, the lack of details regarding diagnosis might give the sense of a homogeneous group of individuals, which this population certainly is not.

Moreover, despite the fact that the Satisfaction with Life Scale is one of the most prominent and commonly used instruments to measure life satisfaction, there are other instruments that can be used in this endeavor. For example, the Life Satisfaction Questionnaire was originally developed as a checklist (Fugl-Meyer, Bränholm & Fugl-Meyer, 1991) targeting important life domains (e.g., vocational, financial, leisure activities, friendships, sexual life, self-care, family, partner relationships) and both physical and psychological health (Melin, Fugl-Meyer & Fugl-Meyer, 2003). Finally, since life satisfaction might be affected by mood, which can be of great variation in some mental-illnesses, it is plausible to suggest that future studies need to address the test–retest reliability of the Satisfaction with Life Scale in this population and to conduct longitudinal studies. That being said, the IRT analyses conducted in this study regarding reliable item information are not influenced by the characteristic of the sample (Oishi, 2007; Kijima et al., 1998). Hence, since the main idea behind IRT is that we can measure the latent trait, in this case life satisfaction, we should expect that analyses of test-retest reliability should yield the same results as here.

Conclusion and last remarks

The psychometric characteristics of the Satisfaction with Life Scale in this sample of Swedish psychiatric patients was acceptable and consistent with previous research suggesting that its five items measure a single factor of life satisfaction. Items 1 (“In most ways my life is close to my ideal”) and 3 (“I am satisfied with my life”) had the two highest discrimination estimates and provided more information than any of the remaining items. That being said, while the Satisfaction with Life Scale had good information and reliability across most of the latent trait range, it could not measure extreme levels of life satisfaction in our sample of Swedish individuals with mental illness. Thus, some modification might be warranted. For instance, item 5 (“If I could live my life over, I would change almost nothing”) had low information value and item 2 (“The conditions of my life are excellent”) seems to have redundant difficulty.

Moreover, we need to consider recent research (Nima et al., 2020a, 2020b) suggesting that life satisfaction might be its own construct, but that it also is part of a general subjective well-being factor. In other words, subjective well-being consists of, besides life satisfaction as a cognitive component, an affective component (i.e., positive and negative affect) and a behavioral component (i.e., harmony in life). These three components are in direct connection to each other and are therefore needed to understand people’s subjective well-being. Thus, only assessing life satisfaction, give us an incomplete picture of how people are actually feeling about their life and an even smaller insight in their well-being as a whole.

That being said, IRT analyses might help clinicians to understand patients’ behavior in relation to the patient’s responses to the items in the test. For example, while some patients are low in their life satisfaction, they might respond with high levels to a specific item in the test. This specific item might, in turn, give important cues for intervention (Pires et al., 2013). This might be essential; after all, in order to address mental illness in the 21st century, we need to develop methods for the prevention, identification and treatment of mental illness; but also, develop methods for the promotion, identification, and maintenance of well-being (see for example Cloninger et al., 2019; Granjard et al., 2021a).

Supplemental Information

Supplemental Information 1 Raw data.

The responses to both demographics and to the Satisfaction with Life Scale.

Click here for additional data file.

We would like to thank the participants and the staff at the five Municipalities in Blekinge for making this research possible. We would also like to thank Ledningssamverkan vård och omsorg (LSVO) in Blekinge for allowing this research. Last but not the least, we want to express our gratitude to Lars-Henry Gustle, Ulrika Harris, and Carina Ström from FoU-avtalet, who designed and conducted the data collection together with other colleagues at the Centre of Competence, Region Blekinge.

Additional Information and Declarations

Competing Interests

Author Contributions

Human Ethics

Data Availability

The authors declare that they have no competing interests. Danilo Garcia is the Head of Research of the Blekinge Center of Competence, which is the Region Blekinge’s research and development unit. The Center works on innovations in public health and practice through interdisciplinary scientific research, community projects, and the dissemination of knowledge in order to increase the quality of life of the habitants of the county of Blekinge, Sweden. Danilo Garcia is one of the founders and the leading researcher of the Network for Well-Being—A network of researchers and students interested in the Science of Well-being.

Danilo Garcia conceived and designed the experiments, performed the experiments, analyzed the data, prepared figures and/or tables, authored or reviewed drafts of the paper, and approved the final draft.

Ali Al Nima conceived and designed the experiments, performed the experiments, analyzed the data, prepared figures and/or tables, authored or reviewed drafts of the paper, and approved the final draft.

Maryam Kazemitabar performed the experiments, analyzed the data, prepared figures and/or tables, authored or reviewed drafts of the paper, and approved the final draft.

Clara Amato conceived and designed the experiments, performed the experiments, authored or reviewed drafts of the paper, and approved the final draft.

Franco Lucchese performed the experiments, authored or reviewed drafts of the paper, and approved the final draft.

Marko Mihailovic performed the experiments, authored or reviewed drafts of the paper, and approved the final draft.

Nobuhiko Kijima performed the experiments, authored or reviewed drafts of the paper, and approved the final draft.

The following information was supplied relating to ethical approvals (i.e., approving body and any reference numbers):

The study (protocol 2017/7) was approved by the Swedish Ethical Review Authority (Dnr. 2017/305).

The following information was supplied regarding data availability:

Data are available in the Supplemental File.

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
