# Peer review of "Psychometric properties of the Swedish version of the satisfaction with life scale in a sample of individuals with mental illness"

_PeerJ, doi:10.7717/peerj.11432_

## Round 0.1 · original submission · Major Revisions

The reviewers have found scientific merit in your work, but there are some issues which you should address in a revised version of the text. Please, see their reports so as to have more information about it.

Reviewer 1 ·

Basic reporting

This is an important and interesting study and I appreciate how you choose to focus on the salutogenesis and not the pathogenesis when working on modern management of mental illnesses. You use a clear professional English language and literature references. You background puts the paper in context.

Experimental design

Please describe why you have chosen to use the SWLS instead of many other rating scales for well-being (LiSat for example). On line 83-84 you present SWLS as “for instance” which gives me the feeling that it was a coincidence that you chose this particular rating scale. Could you argue shortly why this scale should be used, for example as one of the most commonly used or that it is easily available to use in clinical situations. Something that tells me why this of all scales should be evaluated.
I think you have argued and described the difference between CTT and IRT, but I miss a discussion about test-retest evaluation. You have chosen a population with mental illness such as depression, bipolar disease or personality disorders. Test-retest reliability would be very interesting in this population since well-being can be affected by mood and affections which can be of great variation in some mental- illnesses. Is it stable to use at any time or could it be different result using it a week later?
How did the survey take place? A postal survey or at the out patient clinic? Did they fill it in by themselves or with help of others?
What were the different diagnosis in the population? Mental-illness is a very wide concept and a more specific details would be preferable. Where there affective disorders, psychosis, personality disorder. I would like to see a bit more detailed description of the population. For example no one would accept a description of “neurological disease” when evaluation a rating scale, it would be needed to specify for example Multiple sclerosis or chronic neurological degenerative disorders. Mental illness is very wide and it does not make the patients right to not discriminate this better. I also think that the scale could differ in different mental disorders so I would like to see a more described population.

Validity of the findings

There are a lot of numbers and percentages in the participants-section and the reading does not enables of these numbers but instead are a bit difficult. Could the population be described in a table?

The findings are well presented and argued for. What I miss is an very important part of psychometric properties, the data completeness. if there are a lot if missing data it suggests that the participant does not understand the questionnaire or are not able to use it properly. I miss comments on missing data, I doubt that over 200 individuals actually filled in every item adequately, there usually are some missing data.

As mentioned before I also miss test-retest reliability in this particular population, or have the population described better. You can also describe and argue why no test-rest was made and that could be sufficient, but looking from a clinical perspective I argue that both data completeness and test-rest evaluation are more important than some of your result. If the result differs from one week to another, is it then life satisfaction that are measured or something else? and if a lot of data are missing can you then rely on the scale? This could be discussed in the limitation section or for future research.

Additional comments

Taken together, it is an important study but I miss some details in the methodology and description and presentation of the population. I think that by only name the population “ a sample of individuals with mental illness” you minimizes the patient group as packing them together in a very broad concept. I suggest a revision as described above and I think this could make the article more appealing for the reader and it could strengthen its credibility.

·

Basic reporting

The authors present an English well-written study, with a robust analysis of the psychometric properties of the Swedish-adapted version of the life satisfaction scale in a sample of individuals with mental illness.
The methods are adequate and consistent with the objective of the research that has been clearly stated in the introduction.
The tables and figures presented provided detailed information about the psychometric study.

Experimental design

The research question is clearly defined and aligned with the aims of the journal.
This study presents a robust methodology, and a well-performed statistical analyses, that constitutes a strength of the manuscript from my point of view.

Validity of the findings

The authors have constructed an introduction to put the objective of this research into the background. It is clearly identified that this instrument has not been used previously for the purposes that are intended to be explored: measuring the satisfaction with life in a population with mental illness, with or without occupation.
Conclusions are well supported by the analyses and are well connected with the research question.
This will allow mental health professionals in Sweden to use this tool in a standardized way with their patients. It would be interesting to continue researching to provide thresholds of interpretation or classification that improves their use in the clinic, such as minimal clinically important change, ...

Additional comments

The authors present a well-written study, with a robust analysis of the psychometric properties of the Swedish-adapted version of the life satisfaction scale in a sample of individuals with mental illness.
I would like to congratulate the authors on the work. The factor validity analyses, especially in the confirmatory and invariance study that are presented, are robust and methodologically well-executed, which generate interesting conclusions for the practical application of this scale in this study population. The tables and figures presented provided detailed information about the psychometric study.
Please, find below some comments and questions about the manuscript that have arisen after careful reading, and whose objective is to contribute to the improvement of the manuscript readability:

• In the method section lines 165-169… you have used the “listwise method to have the missing data”. Could you provide readers more information about this process?
- Could you provide information about the sample size used to perform this analysis?. Is there a methodological justification? Is the sample size sufficient to support the analyzes carried out? Thus, describe it in the text to give more power to the results.
• Line 178. The value of Cronbach’s Alpha could perhaps be placed in the results section, rather than in the method.
• Line 180, Statistical Procedure. When you apply exploratory factor analysis using principal component analysis with SPSS, could you indicate if you have used any type of rotation (Varimax, etc.) in the process?
• Line 199, Results-Exploratory factor analysis. Can you indicate in the text the percentage of the variance explained by the one-dimensional solution?
• When you write that only one factor with eigenvalues greater than 1 was found, perhaps they should explain this factor extraction condition previously in Methods.
• Line 222 and 232. Use of abbreviations. Define CFIs, before using it in the text for the first time.
• Line 237-238. When you describe the discrimination values (from 1.95 to 3.81), could you indicate which specific parameter/index you are referring to?

Thank you very much,
Sincerely

Reviewer 3 ·

Basic reporting

The text is generally well structured and contains valuable information. However, I have some concerns of minor details, and remarks on references. However, I also have one major reflection concerning the concept of life satisfaction in relation to the dichotomous aspects of pathology and health. The authors could be clearer in the introduction section in defining what the instrument SWLS is measuring. This can clarify the concept life satisfaction in relation to subject well-being and quality of life.

My considerations:
Abstract:
• Line 29: The reference to WHO (2001) is accurate in relation to mental health, but the original reference should be WHO (1948). The WHO (2001) reference is also given in the introduction, and as this reference concerns a shift from focusing disease to illness, or a medical to salutogenic approach, it seems vital to the current context in measuring life satisfaction. In Ottawa Charter (1986) this aspect of health is further developed and could be considered for reference inclusion in the introduction section.
• Line 39-40: I find it unnecessary to include the measurement information for SWLS in the abstract section.

Introduction:
I suggest that the text should be scrutinized and changed in some sentences. In the introduction section, e.g. (Line 68) “In Sweden, approximately 50% of people who are on sick leave are on sick leave due to mental illness.” Suggestion: About half of the proportion on sick leave in Sweden are due to mental illness. (Line 108) “….weigh more or less in relation to …”. Suggestion: …. might be of varying weigh in relation to …”.
(Line 77): Consider alternative or complement to the reference WHO (2001), also given in abstract. The WHO (2001) refer to WHO (1946) (From reference WHO (2001), page 3:“The importance of mental health has been recognized by WHO since its origin, and is reflected by the definition of health in the WHO Constitution as….).

Some considerations regarding references:

Article or chapter by Pavot (2018), and articles by Jovanovic et al (2020), and Pires et al (2013), are missing in the reference list.
The articles Nima et al (2020 a and b) are presented in the text with different publication years (2020ab and 2021), or is there a missing reference?
In the text references for Lee (2014), and Sachs (2004), are presented in the reference list with other publication years (Lee (2018), and Sachs (2003)).
There are different publication years for Kline, R. B. (2005 in the text, and 2018 in reference list (different editions?)). The same difference is noticed regarding Tabachnick and Fidell (in the text 1996, and in the list 2012).

Experimental design

The design and statistical methods is clearly described and suitable for the study. Some minor remarks:
Line 163: I suggest that the foot note is implemented in running text.
Line 165: Replacing the term “drugs disorder” with substance use disorder.

Validity of the findings

I have no concerns regarding the well-structured and explained statistics and results. The tables and figures give complementary information to the text. Raw data has been reviewed and found accurate.
Conclusions and suggestions for further research seems well established.

Additional comments

Thank you for the commendable analyse of this highly used instrument in quality of life research. The results will be, as you have pointed out, of great value in the future research.
I have just a few remarks in my review but ask you to correct the references

---

## Round 0.2 · accepted · Accept

All the reviewers' concerns have been correctly addressed.

Reviewer 1 ·

Basic reporting

I have no further questions or comments.

Experimental design

I have no further questions or comments.

Validity of the findings

I have no further questions or comments.

Additional comments

The authors have responded very well to my comments and have added supplementing details in the article adequately. The inquires that could not be fulfilled are well argued for and/or commented as a limitation. I think this is a well-written and interesting article on an important topic. After the revision it has a better credebility and I think it is ready to be published.

·

Basic reporting

The authors have carried out a systematic validation process in accordance with the principles of the COSMIN statement. It is well written, with a clear and coherent structure.

Experimental design

The design of the work complies with the technical and statistical standards of the validation studies. Confirmatory factor analysis has been included in the validity study, which gives it special interest.
All analyses have been correctly and completely described in the methodology section.

Validity of the findings

The conclusions of the work correspond to the objectives set in the work. This study will allow mental health professionals in Sweden to use this tool in a standardized way. It would be interesting to continue this research line to provide thresholds for the score interpretation or patient classification that improves their use in the clinic, such as minimal clinically important change, or Patient Acceptable Symptom State.

Additional comments

The authors have adequately responded to these reviewers' comments. In my opinion, the manuscript has improved its readability and quality in the aspects indicated in the first review.
Congratulations.